# The Impact of Cardiac Chamber Volumes on Permanent His Bundle Pacing Procedural Outcomes—A Single Center Experience

**DOI:** 10.3390/jcm11237076

**Published:** 2022-11-29

**Authors:** Catalin Pestrea, Ecaterina Cicala, Madalina Ivascu, Alexandra Gherghina, Irina Pintilie, Florin Ortan, Dana Pop

**Affiliations:** 1Department of Interventional Cardiology, Brasov County Clinical Emergency Hospital, 500326 Brasov, Romania; 25th Department of Internal Medicine, Faculty of Medicine, “Iuliu Hațieganu” University of Medicine and Pharmacy, 400012 Cluj-Napoca, Romania; 3Department of Cardiology, Clinical Rehabilitation Hospital, 400347 Cluj-Napoca, Romania

**Keywords:** His bundle electrogram, physiological pacing, cardiac chamber volume, non-deflectable catheter

## Abstract

His bundle pacing (HBP) has several pitfalls, such as the inability to identify the His bundle and lack of capture at acceptable thresholds. The majority of data regarding HBP were obtained using a dedicated non-deflectable delivery system. This study aimed to evaluate the impact of cardiac chamber dimensions on permanent HBP procedural outcomes when using this type of fixed-curve catheter. Seventy-two patients subjected to HBP from the 1st of January to the 31st of December 2021 at our institution were retrospectively reviewed. The baseline clinical characteristics and echocardiographic measurements of all the cardiac chambers were recorded, as well as procedural outcomes (HB electrogram identification and overall procedural success). During the procedure, the HB electrogram was recorded in 59 patients (81.9%) and successful permanent HBP was achieved in 33 patients, representing 45.8% of all the studied patients. Left atrial (LA) and right atrial (RA) volumes were significantly higher in patients without HB electrogram identification. Only LA and RA volumes were statistically associated with HB electrogram localization, while there was no significant association between the echocardiographic parameters and procedural success. LA volumes above 93 mL and RA volumes above 60 mL had an 8.81 times higher chance of failure to localize the HB electrogram compared with patients with lower volumes (*p* < 0.001). When considering non-deflectable delivery catheters for HBP, careful preprocedural echocardiographic analysis of the atrial volumes could help in the proper selection of implanting tools, thus optimizing the procedural outcomes and costs.

## 1. Introduction

Permanent His bundle pacing (HBP) is a physiological pacing technique using bundle branches for both rapid and synchronous biventricular activation. The goal of the procedure is to place the tip of the pacing lead in the His bundle area and to ensure adequate capture of the conduction system [1]. Procedural success increased significantly once a dedicated delivery system for HBP was introduced more than a decade ago. This system included a fixed-curved catheter (Medtronic C315 His, Medtronic, Minneapolis, MN, USA) and a 4.1 Fr lumenless lead with an exposed helix (Medtronic Select Secure 3830, Medtronic, Minneapolis, MN, USA) [2]. The vast majority of studies evaluating the feasibility of the procedure were performed using these dedicated tools [2]. The procedural success rates reported in different centers vary between 75 and 85% [3,4]. The main causes for failure were the inability to find the His bundle during the procedure or, once found, failure to capture at reasonable pacing thresholds.

The anatomy of the heart influences the outcomes of permanent HBP, especially if a non-deflectable catheter is used, but the exact association between each cardiac chamber anatomy and the final result is less known.

The purpose of this study was to evaluate the impact of cardiac chamber volumes on both the ability to locate the His bundle electrogram (HBE) during the procedure and the final procedural success when using the non-deflectable Medtronic C315His catheter and the Medtronic SelectSecure 3830 lead.

## 2. Materials and Methods

### 2.1. Study Design

This was a retrospective, analytical, single-center study.

### 2.2. Patient Selection

A total of 122 consecutive patients who underwent a physiological pacing procedure (with HBP as the first option), either for bradyarrhythmias or for cardiac resynchronization therapy, between 1 January and 31 December 2021 in the Cardiac Pacing Laboratory of the Brașov County Clinical Emergency Hospital in Romania were reviewed for inclusion in the study. The inclusion criteria were the availability of intraprocedural electrocardiograms to document the success of His bundle identification and capture and complete preprocedural echocardiographic measurements of cardiac chambers dimensions. In the end, 72 patients were included in the analysis.

The baseline demographic and clinical characteristics of the patients were recorded.

### 2.3. Pacing Procedure

The procedures were performed by an experienced physician with more than one hundred physiological pacing procedures per year.

The physiological pacing procedure at our institution was performed as follows: the C315 His catheter was first placed at the atrioventricular junction and the Select Secure 3830 lead was advanced only to expose the helix. Unipolar mapping was performed to identify the HBE. In case of initial failure, manual reshaping of the catheter was performed at times according to the operator`s decision. The intracardiac electrograms were recorded using the Workmate Claris EP system (Abbott Cardiovascular, Plymouth, MN, USA). If, after a maximum of ten minutes of fluoroscopy, the HBE could not be found or if the HBE was identified but adequate capture with a threshold less than 2 volts at 1-millisecond pulse duration was not achieved after three attempts, the HBP procedure was abandoned, and left bundle branch area pacing was further attempted. The successful capture of the His bundle was defined as a paced QRS duration of less than 130 ms for patients with narrow baseline QRS (either selective or non-selective pacing) and a reduction in QRS duration to less than 130 ms for patients with wide baseline QRS complex.

The outcome of locating and capturing the His bundle was noted, as well as the pacing and sensing thresholds and the fluoroscopy and total procedural times.

### 2.4. Echocardiographic Parameters

The echocardiographic studies were performed by two experienced physicians using the same echocardiography machine. All the patients included in the study were evaluated before the procedure and the following parameters were recorded using the methods recommended by the current guidelines [5]: the left atrial (LA) volume using the disk summation algorithm in both apical four and two-chamber views, the right atrial (RA) volume using the single-plane method of disks in the apical four-chamber view, the end-systolic and end-diastolic volumes of the left ventricle using the tracings of the blood-tissue interface in the apical four- and apical two-chamber views, and for the right ventricle dimension, the apical four-chamber view using the maximal transversal dimension in the basal one-third of RV inflow at end-diastole. The ejection fraction was assessed using the biplane method of disks (modified Simpson’s rule), and the tricuspid annular plane systolic excursion (TAPSE) was measured in M-mode with the alignment of the cursor with the direction of right ventricular longitudinal excursion from the apical four-chamber view. Mitral and tricuspid regurgitations were quantified using the color flow regurgitant jet and the vena contracta width.

### 2.5. Follow-Up

The patients with successful HBP were followed for a period of six months. Pacing and sensing thresholds and lead or procedure-related complications were recorded at the end of the follow-up period.

### 2.6. Statistical Analysis

Continuous variables are presented as the mean ± one standard deviation or as the median and interquartile range. Categorical variables are presented as frequencies and percentages. A statistical comparison of means was performed using the *t*-test or the Mann–Whitney U test for independent groups and the *t*-test or Wilcoxon test for dependent groups according to the normality of distribution. Cut-off values for echocardiographic parameters were identified from the receiver operating characteristic (ROC) curves. Logistic regression was used to evaluate the association between continuous and categorical variables. A confidence interval of 95% was used for all the tests, and a *p* < 0.05 was considered statistically significant.

Statistical analysis was performed using the SPSS software v 26.0 (IBM, Armonk, NY, USA).

### 2.7. Ethical Considerations

The study complied with all aspects of the Declaration of Helsinki and was approved by the institutional ethics committee.

All the patients were informed and provided their written consent before the procedure.

## 3. Results

The baseline characteristics of the patients are presented in Table 1.

### 3.1. Procedural Characteristics

During the physiological pacing procedure, the HBE was identified in 59 patients and successful permanent HBP was achieved in 33 patients. The reason for procedural failure in the latter group was the lack of capture of the His bundle and capture at high thresholds. The procedural characteristics for the patients with successful procedures were as follows: the pacing threshold was 0.84 ± 0.46 V at a 1 ms pulse width, the QRS detection was 4.25 ± 2.54 mV, the fluoroscopy time was 8.08 ± 6.42 min and the total procedural time was 121.45 ± 26.65 min. The paced QRS duration was 111.67 ± 19.33 msec, significantly narrower than the baseline QRS duration (*p* = 0.016).

The six-month follow-up showed constant pacing (0.86 ± 0.44 V at 1 ms pulse width, *p* = 0.68) and sensing parameters (4.33 ± 3.21 mV, *p* = 0.90), without any lead- or procedure-related complications.

### 3.2. Echocardiographic Data

The LA and RA volumes were significantly higher in patients without HBE identification, while all of the other studied echocardiographic parameters were similar in the two groups. On the other hand, there was no difference in any of the echocardiographic parameters between the groups with procedural failure and procedural success. The recorded echocardiographic parameters and the statistical differences between the different groups are shown in Table 2.

To assess the relationship between the echocardiographic parameters and the two procedural outcomes (HBE identification and overall procedural success), logistic regression was performed, adjusted for height and weight (Table 3). Only LA and RA volumes were statistically associated with HBE identification, while there was no significant association between the echocardiographic parameters and procedural success.

To identify the cut-off values of LA and RA volumes for predicting HBE localization failure, receiver operating characteristic curves were created for each parameter (Figure 1).

A cut-off value for the LA volume of 93 mL and a cut-off value for the RA volume of 60 mL with a sensitivity of 69.2% and a specificity of 80% were identified. After grouping the patients according to these values, LA or RA volumes above the mentioned cut-off values were associated with an 8.81 times higher chance of failure to localize the HBE compared with patients with lower volumes (*p* < 0.001).

## 4. Discussion

The main findings of this study were that both right and left atrial volumes were significantly higher in patients with failed HBE identification and that atrial enlargement was significantly associated with this outcome during permanent HBP procedures using a non-deflectable catheter. Possible explanations are an increased distance from the orifice of the superior cava vein to the atrioventricular junction and anterior displacement of the His bundle position, which decreases the angle of approach from above. Both a longer distance and an acute angle to reach the His bundle position cannot be easily compensated by a fixed-curve sheath.

On the other hand, neither ventricular enlargement nor ventricular function was associated with the outcome of locating the HBE. This may be due to the position of the ventricles distal to the target zone and a smaller impact on His bundle trajectory in ventricular enlargement. As supporting data for this finding, two randomized trials evaluating HBP for cardiac resynchronization therapy, compared to biventricular pacing, in patients with dilated cardiomyopathy reported failures to map the His bundle in only 2 out of 21 patients [6] and in only 1 out of 25 patients, respectively [7].

The cut-off atrial volumes identified in this study are not to be considered absolute reference values, but more as proof that severe enlargement of the left atrium compared to only a moderate enlargement of the right atrium impacts the HBP procedure.

None of the echocardiographic parameters measured were associated with final permanent HBP success. Furthermore, a significant percentage of patients with identified HBE failed to complete the procedure. This was either due to difficulty in lead fixation or an inability to capture the His bundle at acceptable thresholds. Factors involved could be a deeper position of the His bundle in the septum, a thick fibrous sheath encasing the bundle, or local fibrosis and degeneration [8]. This finding confirms the pitfalls of permanent HBP described in previous studies and the importance of the electrophysiological properties of the conduction system in each patient. The correction of distal conduction abnormalities below the bundle of His is not achievable with this procedure, whether the goal is to capture the conduction system in narrow QRS patients [9] or to correct the existing bundle branch block [10].

The overall HBP success in this study was close to 50%, lower than most studies in the literature, which reported success rates of around 80% [11]. A possible explanation could be the protocol used at our institution for conduction system pacing, with lower fluoroscopy times allocated for His bundle localization (a maximum of ten minutes), fewer attempts of fixating the lead (maximum of three), and lower acceptable His bundle capture thresholds (less than 2V at 1 msec), before switching to left bundle branch area pacing. Additionally, the study patients were not selected for the procedure based on their anatomy or electrocardiographic aspects. In the end, 54.3% had a baseline bundle branch block morphology and 62.4% had an atrioventricular block. Both of these conditions could be associated with infra-hisian conduction abnormalities, which significantly reduce the chance of adequate His bundle capture. In the two randomized trials mentioned above that studied patients with left bundle branch block, the HBP arms had a success rate of 52% and 72%, respectively [6,7].

As presented in other studies, we also found that once successfully paced, the His bundle had stable pacing and sensing parameters, without procedure or lead-related complications at follow-up [12].

Several limitations of the study should be mentioned, including the small number of patients and the retrospective data analysis. Additionally, the atrial volumes were assessed using 2D echocardiography, whereas 3D echocardiography may have given a more accurate estimate of the real volume [13]. Accurate measurement of the distance from the superior vena cava–RA junction to the tricuspid annulus may be a more suitable parameter than the entire RA volume, but it is more difficult to acquire in practice [14]. Besides cardiac anatomy, anomalies of the venous system with acute angles or kinking may limit the superior reach of the non-deflectable sheath tip.

As was performed in some of our cases, experienced operators manually modify the curvature of the catheter to accommodate challenging anatomies [14], but with the recent introduction in clinical practice of the deflectable conduction system pacing catheters, many of the difficulties presented in this study may no longer be an issue. Nevertheless, these new catheters are more expensive, require a larger introducer sheath (9F vs. 7F), and are stiffer and sometimes more difficult to maneuver.

The key message of this study is that echocardiographic examination before the procedure, with an emphasis on atrial volumes, may aid physicians who are at the beginning of their learning curve in the proper selection of delivery catheters to increase the chance of HBE identification, especially when the use of different catheters is problematic due to limited availability or cost.

## 5. Conclusions

When considering non-deflectable delivery catheters for HBP, both right and left atrial enlargement are associated with a significant risk of failure to identify the HBE during the procedure but without a significant impact on overall procedural success.

## Figures and Tables

**Figure 1 jcm-11-07076-f001:**
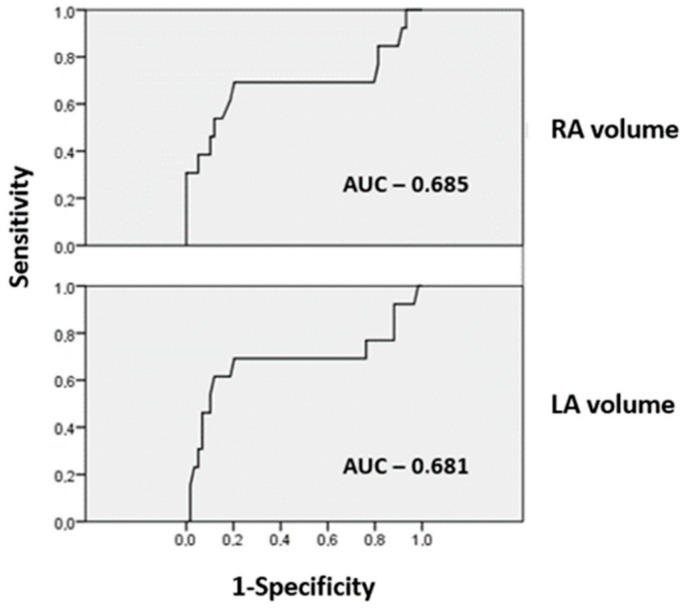
Receiver operating characteristic curve to assess the accuracy of RA and LA volumes in predicting HBE localization during the procedure. LA—left atrium; RA—right atrium; AUC—area under the curve.

**Table 1 jcm-11-07076-t001:** Baseline patient characteristics.

N	72
Age (years, median (Q1–Q3))	70 (65–77)
Male sex	53 (73.6%)
Height (cm, median (Q1–Q3))	172 (165–176.75)
Weight (kg, median (Q1–Q3))	80 (73.25–90)
BMI (kg/cm^2^, median (Q1–Q3))	27,72 (25.17–32.43)
Pacing indication	
Sick sinus node disease	3 (4.2%)
Atrioventricular block	45 (62.4%)
Resynchronization therapy	11 (15.3%)
Pace and ablate	12 (16.7%)
Pacing-induced cardiomyopathy	1 (1.4%)
QRS duration (msec, median (Q1–Q3))	131 (100–160)
Normal QRS	33 (45.8%)
Left bundle branch block	23 (31.9%)
Right bundle branch block	16 (22.2%)
Comorbidities	
Atrial fibrillation	29 (40.3%)
Stroke	6 (8.3%)
Diabetes mellitus	21 (29.2%)
Hypertension	57 (79.2%)
Ischemic cardiomyopathy	22 (30.6%)
Renal insufficiency	18 (25%)
Obstructive lung disease	5 (6.9%)
Medical treatment	
Beta-blockers	56 (77.8%)
Angiotensin-converting enzyme inhibitors	56 (77.8%)
Mineralocorticoid receptor antagonists	29 (40.3%)
ARNI	9 (12.5%)
iSGLT2	2 (2.8%)
Anticoagulants	31 (43%)

Q1–Q3—interquartile range; BMI—body mass index; ARNI—angiotensin receptor-neprilysin inhibitor; iSGLT2—sodium-glucose cotransporter-2 inhibitors.

**Table 2 jcm-11-07076-t002:** Echocardiographic parameters for different study groups.

	HBE Identification			Procedural Success		
	YES	NO	*p*-Value	YES	NO	*p*-Value
N	59	13		33	39	
LA volume (mL, mean ± SD)	73.5 ± 33.1	106.3 ± 47.2	0.043	69.8 ± 23.3	87.6 ± 45.5	0.218
RA volume (mL, mean ± SD)	48.2 ± 17	69.2 ± 32.7	0.038	49.8 ± 18.1	53.8 ± 24.9	0.635
LVES volume (mL, mean ± SD)	69.6 ± 46.2	70.7 ± 34.4	0.578	65.1 ± 41	73.7 ± 46.7	0.403
LVED volume (mL, mean ± SD)	122.9 ± 51.9	122.7 ± 43	0.977	116.4 ± 42	128.3 ± 56.1	0.429
RV basal diameter (mm, mean ± SD)	31.4 ± 4.5	33.5 ± 3.6	0.115	30.7 ± 4.8	32.7 ± 3.7	0.062
Ejection fraction (%, mean ± SD)	48.8 ± 16.2	42.9 ± 14.3	0.139	49.7 ± 15.5	46.1 ± 16.2	0.222
TAPSE (mm, mean ± SD)	20.4 ± 4.6	17.6 ± 5.8	0.063	19.7 ± 4.9	20 ± 4.9	0.928
Mitral regurgitation (mean ± SD)	2.1 ± 0.8	2.4 ± 0.9	0.214	2.2 ± 0.8	2.1 ± 0.9	0.785
Tricuspid regurgitation (mean ± SD)	1.7 ± 0.8	1.9 ± 0.9	0.454	1.8 ± 0.8	1.7 ± 0.7	0.694

LA—left atrium; RA—right atrium; LVES—left ventricular end-systolic; LVED—left ventricular end-diastolic; RV—right ventricle; TAPSE—tricuspid annular plane systolic excursion; SD—standard deviation.

**Table 3 jcm-11-07076-t003:** Logistic regression analysis adjusted for weight and height for procedural outcomes.

	HBE Identification Failure			Overall Procedural Failure		
	OR	95% C.I.	*p*-Value	OR	95% C.I.	*p*-Value
LA volume (mL)	1.021	1.005–1.037	0.010	1.013	0.998–1.029	0.083
RA volume (mL)	1.047	1.013–1.082	0.006	1.005	0.982–1.028	0.689
LVES volume (mL)	1.002	0.987–1.016	0.813	1.003	0.991–1.015	0.629
LVED volume (mL)	1.001	0.987–1.014	0.914	1.003	0.992–1.014	0.603
RV basal diameter (mm)	1.168	0.985–1386	0.075	1.106	0.983–1.244	0.093
Ejection fraction (%)	0.975	0.939–1.012	0.184	0.985	0.955–1.016	0.336
TAPSE (mm)	0.891	0.783–1.014	0.081	1.010	0.917–1.112	0.840
Mitral regurgitation	1.523	0.746–3.109	0.248	0.985	0.562–1.728	0.959
Tricuspid regurgitation	1.321	0.603–2.894	0.487	0.910	0.488–1.696	0.766
Height (cm)	0.965	0.884–1.054	0.428	1.039	0.971–1.112	0.265
Weight (kg)	1.033	0.984–1.084	0.195	1.011	0.973–1.050	0.585

LA—left atrium; RA—right atrium; LVES—left ventricular end-systolic; LVED—left ventricular end-diastolic; RV—right ventricle; TAPSE—tricuspid annular plane systolic excursion; OR—odds ratio; C.I.—confidence interval.

## Data Availability

The datasets are available upon reasonable request to the corresponding author.

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
