# Peer review of "The Impact of Cardiac Chamber Volumes on Permanent His Bundle Pacing Procedural Outcomes—A Single Center Experience"

_jcm, 2022, doi:10.3390/jcm11237076_

Round 1

Reviewer 1 Report

The authors performed a retrospective single centre study evaluating predictors of successful permanent his bundle pacing in 72 patients over one year. Successful his bundle pacing occurred in less than 50%. Authors used a fixed-curve sheath from Medtronic. While large atrial volumes (both LA and RA) were independently predictors of successful HIS bundle localization, no parameter was significantly associated with procedural success (i.e., his bundle pacing).

The authors should focus on one endpoint. In my opinion, successful his bundle pacing is more important than only identification. If the data suggests that LA/RA volumes were not associated with successful his bundle capture, this should be the conclusion of the study. On the contrary, the authors suggest that atrial volumes may play a role in successful his bundle pacing. Apart from this issue, the manuscript is well written with adequate methods and citations.

Minor issues:

1.     The majority of data regarding HB pacing was gained using a dedicated non-deflectable delivery system”. This sentence should be backed up with evidence/literature.

2.     Abstract: There authors should optimize the results section of the abstract. “Only RA and RA volumes were statistically associated with HB electrogram localization, while there was no significant association between the echocardiographic parameters and procedural success”. This sentence repeats the previous sentence that already shows that echocardiographic parameters and procedural success were not associated with each other.

3.     The authors write that the investigator performs more than 100 physiological pacing procedures per year. However, in the analysis covering one year, only 72 patients were included. How many patients receiving physiological pacing at the centre were not included in this analysis?

4.     Fluoroscopy and procedure times, as well as dose area product would be very interesting.

5.     If the authors included height and weight of patients into multivariate analysis, this should be included in Table 3.

6.     The key message of this study is that echocardiographic examination before the procedure, with an emphasis on atrial volumes, may aid physicians who are at the beginning of their learning curve in the proper selection of delivery catheters for His bundle pacing […]”: Really? There was no difference in successful HB pacing in patients with large vs. small atria.

Reviewer 2 Report

Dear Authors 

Your results are very important for the patient's selection preimplantation. A small effort (doing an echocardiographic study with particular attention to the atrial dimensions) can give great results in terms of better patient selection for physiological stimulation.

1. You repeat 12-fold "His bundle pacing". Could you use an abbreviation? HBP

2. Could you better describe the study methods, in particular patients’ selection?

3. 86,87: In this site you can put an abbreviation (not in line 131).

4. 118,119: I would avoid putting percentages relative to the total number of procedures because it could generate confusion. If you enrolled only patients in whom the HB was visible, the results should only be given for this patients group.

5. 131: This is already abbreviated. Would you change with the abbreviation? This problem is present in other positions in the text of the article.

6. 136: Presented? Showed?

7. Discussion: put the abbreviations. The discussion isn't very smooth. I think English needs to be improved.
